# Heavy-tailed Representations, Text Polarity Classification & Data Augmentation

**Hamid Jalalzai**[*]
LTCI, Télécom Paris
Institut Polytechnique de Paris
hamid.jalalzai@telecom-paris.fr

**Pierre Colombo**[*]
IBM France
LTCI, Télécom Paris
Institut Polytechnique de Paris
pierre.colombo@telecom-paris.fr

**Chloé Clavel**
LTCI, Télécom Paris
Institut Polytechnique de Paris
chloe.clavel@telecom-paris.fr

**Eric Gaussier**
Univ. Grenoble Alpes, CNRS, Grenoble INP, LIG
eric.gaussier@imag.fr

**Giovanna Varni**
LTCI, Télécom Paris
Institut Polytechnique de Paris
giovanna.varni@telecom-paris.fr

**Emmanuel Vignon**
IBM France
emmanuel.vignon@fr.ibm.com

**Anne Sabourin**
LTCI, Télécom Paris
Institut Polytechnique de Paris
anne.sabourin@telecom-paris.fr

## Abstract

The dominant approaches to text representation in natural language rely on learning embeddings on massive corpora which have convenient properties such as compositionality and distance preservation. In this paper, we develop a novel method to learn a heavy-tailed embedding with desirable regularity properties regarding the distributional tails, which allows to analyze the points far away from the distribution bulk using the framework of multivariate extreme value theory. In particular, a classifier dedicated to the tails of the proposed embedding is obtained which exhibits a *scale invariance* property exploited in a novel text generation method for label preserving dataset augmentation. Experiments on synthetic and real text data show the relevance of the proposed framework and confirm that this method generates meaningful sentences with controllable attributes, *e.g.* positive or negative sentiments.

## 1 Introduction

Representing the meaning of natural language in a mathematically grounded way is a scientific challenge that has received increasing attention with the explosion of digital content and text data in the last decade. Relying on the richness of contents, several embeddings have been proposed [44, 45, 19] with demonstrated efficiency for the considered tasks when learnt on massive datasets.

---

[*]Both authors contributed equally

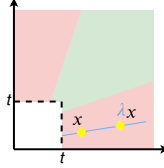

Figure 1: Illustration of angular classifier $g$ dedicated to extremes $\{x, \|x\|_\infty \geq t\}$ in $\mathbb{R}^2_+$. The red and green truncated cones are respectively labeled as $+1$ and $-1$ by $g$.

However, none of these embeddings take into account the fact that word frequency distributions are heavy tailed [2, 11, 40], so that extremes are naturally present in texts (see also Fig. 6a and 6b in the supplementary material). Similarly, [3] shows that, contrary to image taxonomies, the underlying distributions for words and documents in large scale textual taxonomies are also heavy tailed. Exploiting this information, several studies, as [13, 38], were able to improve text mining applications by accurately modeling the tails of textual elements.

In this work, we rely on the framework of multivariate extreme value analysis, based on extreme value theory (EVT) which focuses on the distributional tails. EVT is valid under a regularity assumption which amounts to a homogeneity property above large thresholds: the tail behavior of the considered variables must be well approximated by a power law, see Section 2 for a rigorous statement. The tail region (where samples are considered as extreme) of the input variable $x \in \mathbb{R}^d$ is of the kind $\{\|x\| \geq t\}$, for a large threshold $t$. The latter is typically chosen such that a small but non negligible proportion of the data is considered as extreme, namely $25\%$ in our experiments. A major advantage of this framework in the case of labeled data [30] is that classification on the tail regions may be performed using the angle $\Theta(x) = \|x\|^{-1}x$ only, see Figure 1. The main idea behind the present paper is to take advantage of the scale invariance for two tasks regarding sentiment analysis of text data: *(i)* Improved classification of extreme inputs, *(ii)* Label preserving data augmentation, as the most probable label of an input $x$ is unchanged by multiplying $x$ by $\lambda > 1$.

EVT in a machine learning framework has received increasing attention in the past few years. Learning tasks considered so far include anomaly detection [48, 49, 12, 23, 53], anomaly clustering [9], unsupervised learning [22], online learning [6, 1], dimension reduction and support identification [24, 8, 10, 29]. The present paper builds upon the methodological framework proposed by Jalalzai et al. [30] for classification in extreme regions. The goal of Jalalzai et al. [30] is to improve the performance of classifiers $\widehat{g}(x)$ issued from Empirical Risk Minimization (ERM) on the tail regions $\{\|x\| > t\}$ Indeed, they argue that for very large $t$, there is no guarantee that $\widehat{g}$ would perform well conditionally to $\{\|X\| > t\}$, precisely because of the scarcity of such examples in the training set. They thus propose to train a specific classifier dedicated to extremes leveraging the probabilistic structure of the tails. Jalalzai et al. [30] demonstrate the usefulness of their framework with simulated and some real world datasets. However, there is no reason to assume that the previously mentioned text embeddings satisfy the required regularity assumptions. The aim of the present work is to extend [30]'s methodology to datasets which do not satisfy their assumptions, in particular to text datasets embedded by state of the art techniques. This is achieved by the algorithm *Learning a Heavy Tailed Representation* (in short **LHTR**) which learns a transformation mapping the input data $X$ onto a random vector $Z$ which does satisfy the aforementioned assumptions. The transformation is learnt by an adversarial strategy [26].

In Appendix C we propose an interpretation of the extreme nature of an input in both **LHTR** and BERT representations. In a word, these sequences are longer and are more difficult to handle (for next token prediction and classification tasks) than non extreme ones.

Our second contribution is a novel data augmentation mechanism **GENELIEX** which takes advantage of the scale invariance properties of $Z$ to generate synthetic sequences that keep invariant the attribute of the original sequence. Label preserving data augmentation is an effective solution to the data scarcity problem and is an efficient pre-processing step for moderate dimensional datasets [55, 56]. Adapting these methods to NLP problems remains a challenging issue. The problem consists in constructing a transformation $h$ such that for any sample $x$ with label $y(x)$, the generated sample $h(x)$ would remain label consistent: $y\big(h(x)\big) = y(x)$ [46]. The dominant approaches for text data augmentation rely on word level transformations such as synonym replacement, slot filling, swap deletion [56] using external resources such as wordnet [42]. Linguistic based approaches can also be

combined with vectorial representations provided by language models [32]. However, to the best of our knowledge, building a vectorial transformation without using any external linguistic resources remains an open problem. In this work, as the label $y\big(h(x)\big)$ is unknown as soon as $h(x)$ does not belong to the training set, we address this issue by learning both an embedding $\varphi$ and a classifier $g$ satisfying a relaxed version of the problem above mentioned, namely $\forall \lambda \geq 1$

$$g\big(h_\lambda(\varphi(x))\big) = g\big(\varphi(x)\big). \tag{1}$$

For mathematical reasons which will appear clearly in Section 2.2, $h_\lambda$ is chosen as the homothety with scale factor $\lambda$, $h_\lambda(x) = \lambda x$. In this paper, we work with output vectors issued by BERT [19]. BERT and its variants are currently the most widely used language model but we emphasize that the proposed methodology could equally be applied using any other representation as input. BERT embedding does not satisfy the regularity properties required by EVT (see the results from statistical tests performed in Appendix B.5) Besides, there is no reason why a classifier $g$ trained on such embedding would be scale invariant, *i.e.* would satisfy for a given sequence $u$, embedded as $x$, $g(h_\lambda(x)) = g(x) \ \forall \lambda \geq 1$. On the classification task, we demonstrate on two datasets of sentiment analysis that the embedding learnt by **LHTR** on top of BERT is indeed following a heavy-tailed distribution. Besides, a classifier trained on the embedding learnt by **LHTR** outperforms the same classifier trained on BERT. On the dataset augmentation task, quantitative and qualitative experiments demonstrate the ability of **GENELIEX** to generate new sequences while preserving labels.

The rest of this paper is organized as follows. Section 2 introduces the necessary background in multivariate extremes. The methodology we propose is detailed at length in Section 3. Illustrative numerical experiments on both synthetic and real data are gathered in sections 4 and 5. Further comments and experimental results are provided in the supplementary material.

## 2 Background

### 2.1 Extreme values, heavy tails and regular variation

Extreme value analysis is a branch of statistics whose main focus is on events characterized by an unusually high value of a monitored quantity. A convenient working assumption in EVT is *regular variation*. A real-valued random variable $X$ is regularly varying with index $\alpha > 0$, a property denoted as $RV(\alpha)$, if and only if there exists a function $b(t) > 0$, with $b(t) \to \infty$ as $t \to \infty$, such that for any fixed $x > 0$: $t\mathbb{P}\{X/b(t) > x\} \xrightarrow[t\to\infty]{} x^{-\alpha}$. In the multivariate case $X = (X_1, \ldots, X_d) \in \mathbb{R}^d$, it is usually assumed that a preliminary component-wise transformation has been applied so that each margin $X_j$ is $RV(1)$ with $b(t) = t$ and takes only positive values. $X$ is *standard multivariate regularly varying* if there exists a positive Radon measure $\mu$ on $[0, \infty]^d \backslash \{0\}$

$$t\mathbb{P}\left\{t^{-1}X \in A\right\} \xrightarrow[t\to\infty]{} \mu(A), \tag{2}$$

for any Borelian set $A \subset [0, \infty]^d$ which is bounded away from $0$ and such that the limit measure $\mu$ of the boundary $\partial A$ is zero. For a complete introduction to the theory of Regular Variation, the reader may refer to [47]. The measure $\mu$ may be understood as the limit distribution of tail events. In (2), $\mu$ is homogeneous of order $-1$, that is $\mu(tA) = t^{-1}\mu(A)$, $t > 0$, $A \subset [0, \infty]^d \setminus \{0\}$. This scale invariance is key for our purposes, as detailed in Section 2.2. The main idea behind extreme value analysis is to learn relevant features of $\mu$ using the largest available data.

### 2.2 Classification in extreme regions

We now recall the classification setup for extremes as introduced in [30]. Let $(X, Y) \in \mathbb{R}_+^d \times \{-1, 1\}$ be a random pair. Authors of [30] assume standard regular variation for both classes, that is $t\mathbb{P}\{X \in tA \mid Y = \pm 1\} \to \mu_\pm(A)$, where $A$ is as in (2). Let $\| \cdot \|$ be any norm on $\mathbb{R}^d$ and consider the risk of a classifier $g : \mathbb{R}_+^d \to \{\pm 1\}$ above a radial threshold $t$,

$$L_t(g) = \mathbb{P}\left\{Y \neq g(X) \mid \|X\| > t\right\}. \tag{3}$$

The goal is to minimize the asymptotic risk in the extremes $L_\infty(g) = \limsup_{t\to\infty} L_t(g)$. Using the scale invariance property of $\mu$, under additional mild regularity assumptions concerning the regression function, namely uniform convergence to the limit at infinity, one can prove the following result

(see [30], Theorem 1): there exists a classifier $g_\infty^\star$ depending on the pseudo-angle $\Theta(x) = \|x\|^{-1}x$ only, that is $g_\infty^\star(x) = g_\infty^\star(\Theta(x))$, which is asymptotically optimal in terms of classification risk, *i.e.* $L_\infty(g_\infty^\star) = \inf_{g \text{ measurable}} L_\infty(g)$. Notice that for $x \in \mathbb{R}_+^d \setminus \{0\}$, the angle $\Theta(x)$ belongs to the positive orthant of the unit sphere, denoted by $S$ in the sequel. As a consequence, the optimal classifiers on extreme regions are based on indicator functions of truncated cones on the kind $\{\|x\| > t, \Theta(x) \in B\}$, where $B \subset S$, see Figure 1. We emphasize that the labels provided by such a classifier remain unchanged when rescaling the samples by a factor $\lambda \geq 1$ (*i.e.* $g(x) = g(\Theta(x)) = g(\Theta(\lambda x)), \forall x \in \{x, \|x\| \geq t\}$). The angular structure of the optimal classifier $g_\infty^\star$ is the basis for the following ERM strategy using the most extreme points of a dataset. Let $\mathcal{G}_S$ be a class of angular classifiers defined on the sphere $S$ with finite VC dimension $V_{\mathcal{G}_S} < \infty$. By extension, for any $x \in \mathbb{R}_+^d$ and $g \in \mathcal{G}_S$, $g(x) = g(\Theta(x)) \in \{-1, 1\}$. Given $n$ training data $\{(X_i, Y_i)\}_{i=1}^n$ made of *i.i.d* copies of $(X, Y)$, sorting the training observations by decreasing order of magnitude, let $X_{(i)}$ (with corresponding sorted label $Y_{(i)}$) denote the $i$-th order statistic, *i.e.* $\|X_{(1)}\| \geq \ldots \geq \|X_{(n)}\|$. The empirical risk for the $k$ largest observations $\widehat{L}_k(g) = \frac{1}{k}\sum_{i=1}^k \mathbf{1}\{Y_{(i)} \neq g(\Theta(X_{(i)}))\}$ is an empirical version of the risk $L_{t(k)}(g)$ as defined in (3) where $t(k)$ is a $(1 - k/n)$-quantile of the norm, $\mathbb{P}\{\|X\| > t(k)\} = k/n$. Selection of $k$ is a bias-variance compromise, see Appendix B for further discussion. The strategy promoted by [30] is to use $\widehat{g}_k = \arg\min_{g \in \mathcal{G}_S} \widehat{L}_k(g)$, for classification in the extreme region $\{x \in \mathbb{R}_+^d : \|x\| > t(k)\}$. The following result provides guarantees concerning the excess risk of $\widehat{g}_k$ compared with the Bayes risk above level $t = t(k)$, $L_t^\star = \inf_{g \text{ measurable}} L_t(g)$.

**Theorem 1** *([30], Theorem 2) If each class satisfies the regular variation assumption (2), under an additional regularity assumption concerning the regression function $\eta(x) = \mathbb{P}\{Y = +1 \mid x\}$ (see Equation (4) in Appendix B.3), for $\delta \in (0, 1)$, $\forall n \geq 1$, it holds with probability larger than $1 - \delta$ that*

$$L_{t(k)}(\widehat{g}_k) - L_{t(k)}^\star \leq \frac{1}{\sqrt{k}}\left(\sqrt{2(1 - k/n)\log(2/\delta)} + C\sqrt{V_{\mathcal{G}_S}\log(1/\delta)}\right) +$$

$$\frac{1}{k}\left(5 + 2\log(1/\delta) + \sqrt{\log(1/\delta)}(C\sqrt{V_{\mathcal{G}_S}} + \sqrt{2})\right) + \left\{\inf_{g \in \mathcal{G}_S} L_{t(k)}(g) - L_{t(k)}^\star\right\},$$

*where $C$ is a universal constant.*

In the present work we do *not* assume that the baseline representation $X$ for text data satisfies the assumptions of Theorem 1. Instead, our goal is is to render the latter theoretical framework applicable by learning a representation which satisfies the regular variation condition given in (2), hereafter referred as Condition (2) which is the main assumption for Theorem 1 to hold. Our experiments demonstrate empirically that enforcing Condition (2) is enough for our purposes, namely improved classification and label preserving data augmentation, see Appendix B.3 for further discussion.

## 3 Heavy-tailed Text Embeddings

### 3.1 Learning a heavy-tailed representation

We now introduce a novel algorithm *Learning a heavy-tailed representation* (`LHTR`) for text data from high dimensional vectors as issued by pre-trained embeddings such as BERT. The idea behind is to modify the output $X$ of BERT so that classification in the tail regions enjoys the statistical guarantees presented in Section 2, while classification in the bulk (where many training points are available) can still be performed using standard models. Stated otherwise, `LHTR` increases the information carried by the resulting vector $Z = \varphi(X) \in \mathbb{R}^{d'}$ regarding the label $Y$ in the tail regions of $Z$ in order to improve the performance of a downstream classifier. In addition `LHTR` is a building block of the data augmentation algorithm `GENELIEX` detailed in Section 3.2. `LHTR` proceeds by training an encoding function $\varphi$ in such a way that *(i)* the marginal distribution $q(z)$ of the code $Z$ be close to a user-specified heavy tailed target distribution $p$ satisfying the regularity condition (2); and *(ii)* the classification loss of a multilayer perceptron trained on the code $Z$ be small.

A major difference distinguishing `LHTR` from existing auto-encoding schemes is that the target distribution on the latent space is not chosen as a Gaussian distribution but as a heavy-tailed, regularly varying one. A workable example of such a target is provided in our experiments (Section 4). As the Bayes classifier (*i.e.* the optimal one among all possible classifiers) in the extreme region has a

potentially different structure from the Bayes classifier on the bulk (recall from Section 2 that the optimal classifier at infinity depends on the angle $\Theta(x)$ only), **LHTR** trains two different classifiers, $g^{\text{ext}}$ on the extreme region of the latent space on the one hand, and $g^{\text{bulk}}$ on its complementary set on the other hand. Given a high threshold $t$, the extreme region of the latent space is defined as the set $\{z : \|z\| > t\}$. In practice, the threshold $t$ is chosen as an empirical quantile of order $(1 - \kappa)$ (for some small, fixed $\kappa$) of the norm of encoded data $\|Z_i\| = \|\varphi(X_i)\|$. The classifier trained by **LHTR** is thus of the kind $g(z) = g^{\text{ext}}(z)\mathbb{1}\{\|z\| > t\} + g^{\text{bulk}}(z)\mathbb{1}\{\|z\| \leq t\}$. If the downstream task is classification on the whole input space, in the end the bulk classifier $g^{\text{bulk}}$ may be replaced with any other classifier $g'$ trained on the original input data $X$ restricted to the non-extreme samples (*i.e.* $\{X_i, \|\varphi(X_i)\| \leq t\}$). Indeed training $g^{\text{bulk}}$ only serves as an intermediate step to learn an adequate representation $\varphi$.

**Remark 1** Recall from Section 2.2 that the optimal classifier in the extreme region as $t \rightarrow \infty$ depends on the angular component $\theta(x)$ only, or in other words, is scale invariant. One can thus reasonably expect the trained classifier $g^{\text{ext}}(z)$ to enjoy the same property. This scale invariance is indeed verified in our experiments (see Sections 4 and 5) and is the starting point for our data augmentation algorithm in Section 3.2. An alternative strategy would be to train an angular classifier, *i.e.* to impose scale invariance. However in preliminary experiments (not shown here), the resulting classifier was less efficient and we decided against this option in view of the scale invariance and better performance of the unconstrained classifier.

The goal of **LHTR** is to minimize the weighted risk

$$R(\varphi, g^{\text{ext}}, g^{\text{bulk}}) = \rho_1 \mathbb{P}\left\{Y \neq g^{\text{ext}}(Z), \|Z\| \geq t\right\} + \rho_2 \mathbb{P}\left\{Y \neq g^{\text{bulk}}(Z), \|Z\| < t\right\} + \rho_3 \mathfrak{D}(q(z), p(z)),$$

where $Z = \varphi(X)$, $\mathfrak{D}$ is the Jensen-Shannon distance between the heavy tailed target distribution $p$ and the code distribution $q$, and $\rho_1, \rho_2, \rho_3$ are positive weights. Following common practice in the adversarial literature, the Jensen-Shannon distance is approached (up to a constant term) by the empirical proxy $\widehat{L}(q, p) = \sup_{D \in \Gamma} \widehat{L}(q, p, D)$, with $\widehat{L}(q, p, D) = \frac{1}{m}\sum_{i=1}^{m} \log D(Z_i) + \log\left(1 - D(\tilde{Z}_i)\right)$, where $\Gamma$ is a wide class of discriminant functions valued in $[0, 1]$, and where independent samples $Z_i, \tilde{Z}_i$ are respectively sampled from the target distribution and the code distribution $q$. Further details on adversarial learning are provided in Appendix A.1. The classifiers $g^{\text{ext}}, g^{\text{bulk}}$ are of the form $g^{\text{ext}}(z) = 2\mathbb{1}\{C^{\text{ext}}(z) > 1/2\} - 1$, $g^{\text{bulk}}(z) = 2\mathbb{1}\{C^{\text{bulk}}(z) > 1/2\} - 1$ where $C^{\text{ext}}, C^{\text{bulk}}$ are also discriminant functions valued in $[0, 1]$. Following common practice, we shall refer to $C^{\text{ext}}, C^{\text{bulk}}$ as classifiers as well. In the end, **LHTR** solves the following min-max problem $\inf_{C^{\text{ext}}, C^{\text{bulk}}, \varphi} \sup_D \widehat{R}(\varphi, C^{\text{ext}}, C^{\text{bulk}}, D)$ with

$$\widehat{R}(\varphi, C^{\text{ext}}, C^{\text{bulk}}, D) = \frac{\rho_1}{k}\sum_{i=1}^{k} \ell(Y_{(i)}, C^{\text{ext}}(Z_{(i)})) + \frac{\rho_2}{n-k}\sum_{i=k+1}^{n-k} \ell(Y_{(i)}, C^{\text{bulk}}(Z_{(i)})) + \rho_3 \hat{L}(q, p, D),$$

where $\{Z_{(i)} = \varphi(X_{(i)}), i = 1, \ldots, n\}$ are the encoded observations with associated labels $Y_{(i)}$ sorted by decreasing magnitude of $\|Z\|$ (*i.e.* $\|Z_{(1)}\| \geq \cdots \geq \|Z_{(n)}\|$), $k = \lfloor \kappa n \rfloor$ is the number of extreme samples among the $n$ encoded observations and $\ell(y, C(x)) = -(y \log C(x) + (1 - y) \log(1 - C(x)), y \in \{0, 1\}$ is the negative log-likelihood of the discriminant function $C(x) \in (0, 1)$. A summary of **LHTR** and an illustration of its workflow are provided in Appendices A.2 and A.3.

### 3.2 A heavy-tailed representation for dataset augmentation

We now introduce **GENELIEX** (Generating Label Invariant sequences from Extremes), a data augmentation algorithm, which relies on the label invariance property under rescaling of the classifier for the extremes learnt by **LHTR**. **GENELIEX** considers input sentences as sequences and follows the seq2seq approach [52, 16, 7]. It trains a Transformer Decoder [54] $G^{\text{ext}}$ on the extreme regions.

For an input sequence $U = (u_1, \ldots, u_T)$ of length $T$, represented as $X_U$ by BERT with latent code $Z = \varphi(X_U)$ lying in the extreme regions, **GENELIEX** produces, through its decoder $G^{\text{ext}}$ $M$ sequences $U'_j$ where $j \in \{1, \ldots, M\}$. The $M$ decoded sequences correspond to the codes $\{\lambda_j Z, j \in \{1, \ldots, M\}\}$ where $\lambda_j > 1$. To generate sequences, the decoder iteratively takes as input the previously generated word (the first word being a start symbol), updates its internal state, and returns the next word with the highest probability. This process is repeated until either the

decoder generates a stop symbol or the length of the generated sequence reaches the maximum length ($T_{\max}$). To train the decoder $G^{\text{ext}} : \mathbb{R}^{d'} \rightarrow \left[1, \dots, |\mathcal{V}|\right]^{T_{\max}}$ where $\mathcal{V}$ is the vocabulary on the extreme regions, **GENELIEX** requires an additional dataset $\mathcal{D}_{g_n} = (U_1, \dots, U_n)$ (not necessarily labeled) with associated representation *via* BERT $(X_{U,1}, \dots, X_{U,n})$. Learning is carried out by optimising the classical negative log-likelihood of individual tokens $\ell_{gen}$. The latter is defined as $\ell_{gen}\big(U, G^{\text{ext}}(\varphi(X))\big) \overset{\text{def}}{=} \sum_{t=1}^{T_{\max}} \sum_{v \in \mathcal{V}} \mathbb{1}\{u_t = v\} \log\big(p_{v,t}\big)$, where $p_{v,t}$ is the probability predicted by $G^{\text{ext}}$ that the $t^{th}$ word is equal to $v$. A detailed description of the training step of **GENELIEX** is provided in Algorithm 2 in Appendix A.3, see also Appendix A.2 for an illustrative diagram.

**Remark 2** Note that the proposed method only augments data on the extreme regions. A general data augmentation algorithm can be obtained by combining this approach with any other algorithm on the original input data $X$ whose latent code $Z = \varphi(X_U)$ does not lie in the extreme regions.

## 4   Experiments : Classification

In our experiments we work with the infinity norm. The proportion of extreme samples in the training step of **LHTR** is chosen as $\kappa = 1/4$. The threshold $t$ defining the extreme region $\{\|x\| > t\}$ in the test set is $t = \|\tilde{Z}_{(\lfloor \kappa n \rfloor)}\|$ as returned by **LHTR**. We denote by $\mathcal{T}_{\text{test}}$ and $\mathcal{T}_{\text{train}}$ respectively the extreme test and train sets thus defined. Classifiers $C^{\text{bulk}}, C^{\text{ext}}$ involved in **LHTR** are Multi Layer Perceptrons (MLP), see Appendix B.6 for a full description of the architectures.

**Heavy-tailed distribution.** The regularly varying target distribution is chosen as a multivariate logistic distribution with parameter $\delta = 0.9$, refer to Appendix B.4 for details and an illustration with various values of $\delta$. This distribution is widely used in the context of extreme values analysis [10, 53, 23] and differ from the classical logistic distribution.

### 4.1   Toy example: about LHTR

We start with a simple bivariate illustration of the heavy tailed representation learnt by **LHTR**. Our goal is to provide insight on how the learnt mapping $\varphi$ acts on the input space and how the transformation affects the definition of extremes (recall that extreme samples are defined as those samples which norm exceeds an empirical quantile). Labeled samples are simulated from a Gaussian mixture

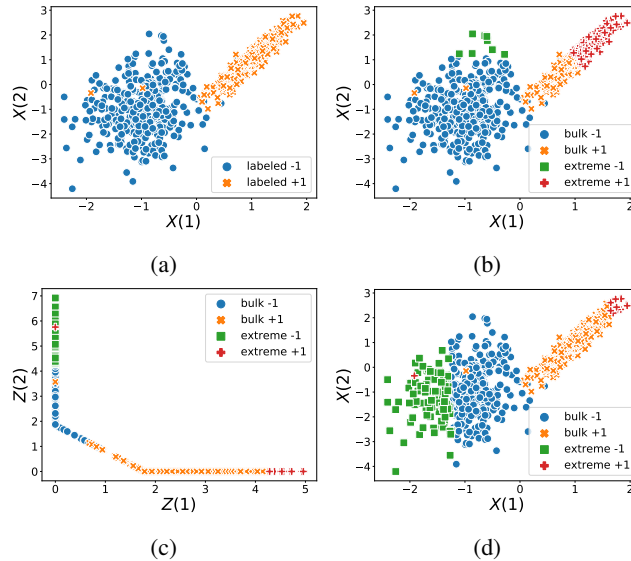

Figure 2: Figure 2a: Bivariate samples $X_i$ in the input space. Figure 2b: $X_i$'s in the input space with extremes from each class selected in the input space. Figure 2c: Latent space representation $Z_i = \varphi(X_i)$. Extremes of each class are selected in the latent space. Figure 2d: $X_i$'s in the input space with extremes from each class selected in the latent space.

distribution with two components of identical weight. The label indicates the component from which

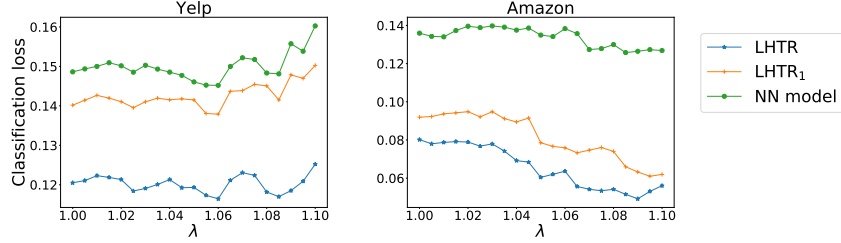

Figure 3: Classification loss of **LHTR**, **LHTR**$_1$ and **NN model** on the extreme test set $\{x \in \mathcal{T}, ||x|| \geq \lambda t\}$ for increasing values of $\lambda$ (X-axis), on *Yelp* and *Amazon*.

the point is generated. **LHTR** is trained on $2250$ examples and a testing set of size $750$ is shown in Figure 2. The testing samples in the input space (Figure 2a) are mapped onto the latent space *via* $\varphi$ (Figure 2c) In Figure 2b, the extreme raw observations are selected according to their norm after a component-wise standardisation of $X_i$, refer to Appendix B for details. The extreme threshold $t$ is chosen as the $75\%$ empirical quantile of the norm on the training set in the input space. Notice in the latter figure the class imbalance among extremes. In Figure 2c, extremes are selected as the $25\%$ samples with the largest norm in the latent space. Figure 2d is similar to Figure 2b except for the selection of extremes which is performed in the latent space as in Figure 2c. On this toy example, the adversarial strategy appears to succeed in learning a code which distribution is close to the logistic target, as illustrated by the similarity between Figure 2c and Figure 5a in the supplementary. In addition, the heavy tailed representation allows a more balanced selection of extremes than the input representation.

## 4.2 Application to positive *vs.* negative classification of sequences

In this section, we dissect **LHTR** to better understand the relative importance of: *(i)* working with a heavy-tailed representation, *(ii)* training two independent classifiers: one dedicated to the bulk and the second one dedicated to the extremes. In addition, we verify experimentally that the latter classifier is scale invariant, which is neither the case for the former, nor for a classifier trained on BERT input.
**Experimental settings.** We compare the performance of three models. The baseline **NN model** is a MLP trained on BERT. The second model **LHTR**$_1$ is a variant of **LHTR** where a single MLP ($C$) is trained on the output of the encoder $\varphi$, using all the available data, both extreme and non extreme ones. The third model (**LHTR**) trains two separate MLP classifiers $C^{\text{ext}}$ and $C^{\text{bulk}}$ respectively dedicated to the extreme and bulk regions of the learnt representation $\varphi$. All models take the same training inputs, use BERT embedding and their classifiers have identical structure, see Appendix A.2 and B.6 for a summary of model workflows and additional details concerning the network architectures.
Comparing **LHTR**$_1$ with **NN model** assesses the relevance of working with heavy-tailed embeddings. Since **LHTR**$_1$ is obtained by using **LHTR** with $C^{\text{ext}} = C^{\text{bulk}}$, comparing **LHTR**$_1$ with **LHTR** validates the use of two separate classifiers so that extremes are handled in a specific manner. As we make no claim concerning the usefulness of **LHTR** in the bulk, at the prediction step we suggest working with a combination of two models: **LHTR** with $C^{ext}$ for extreme samples and any other off-the-shelf ML tool for the remaining samples (*e.g.* **NN model**).
**Datasets.** In our experiments we rely on two large datasets from *Amazon* (231k reviews) [41] and from *Yelp* (1,450k reviews) [58, 36]. Reviews, (made of multiple sentences) with a rating greater than or equal to $4/5$ are labeled as $+1$, while those with a rating smaller or equal to $2/5$ are labeled as $-1$. The gap in reviews' ratings is designed to avoid any overlap between labels of different contents.
**Results.** Figure 3 gathers the results obtained by the three considered classifiers on the tail regions of the two datasets mentioned above. To illustrate the generalization ability of the proposed classifier in the extreme regions we consider nested subsets of the extreme test set $\mathcal{T}_{\text{test}}$, $\mathcal{T}^\lambda = \{z \in \mathcal{T}_{\text{test}}, ||z|| \geq \lambda t\}$, $\lambda \geq 1$. For all factor $\lambda \geq 1$, $\mathcal{T}^\lambda \subseteq \mathcal{T}_{\text{test}}$. The greater $\lambda$, the fewer the samples retained for evaluation and the greater their norms. On both datasets, **LHTR**$_1$ outperforms the baseline **NN model**. This shows the improvement offered by the heavy-tailed embedding on the extreme region. In addition, **LHTR**$_1$ is in turn largely outperformed by the classifier **LHTR**, which proves the importance of working with two separate classifiers. The performance of the proposed model respectively on the bulk region, tail region and overall, is reported in Table 1, which shows that using a specific classifier dedicated to extremes improves the overall performance.

| Model | Amazon | | | Yelp | | |
|---|---|---|---|---|---|---|
| | Bulk | Extreme | Overall | Bulk | Extreme | Overall |
| `NN model` | **0.085** | 0.135 | 0.098 | **0.098** | 0.148 | 0.111 |
| `LHTR`$_1$ | 0.104 | 0.091 | 0.101 | 0.160 | 0.139 | 0.155 |
| `LHTR` | 0.105 | **0.08** | 0.0988 | 0.162 | **0.1205** | 0.152 |
| **Proposed Model** | 0.085 | 0.08 | 0.084 | 0.097 | 0.1205 | 0.103 |

Table 1: Classification losses on *Amazon* and *Yelp*. 'Proposed Model' results from using `NN model` model for the bulk and `LHTR` for the extreme test sets. The extreme region contains 6.9k samples for *Amazon* and 6.1k samples for *Yelp*, both corresponding roughly to 25% of the whole test set size.

**Scale invariance.** On all datasets, the extreme classifier $g^{\text{ext}}$ verifies Equation (1) for each sample of the test set, $g^{\text{ext}}(\lambda Z) = g^{\text{ext}}(Z)$ with $\lambda$ ranging from 1 to 20, demonstrating scale invariance of $g^{\text{ext}}$ on the extreme region. The same experiments conducted both with `NN model` and a MLP classifier trained on BERT and `LHTR`$_1$ show label changes for varying values of $\lambda$: none of them are scale invariant. Appendix B.5 gathers additional experimental details. The scale invariance property will be exploited in the next section to perform label invariant generation.

## 5 Experiments : Label Invariant Generation

### 5.1 Experimental Setting

**Comparison with existing work.** We compare `GENELIEX` with two state of the art methods for dataset augmentation, Wei and Zou [56] and Kobayashi [32]. Contrarily to these works which use heuristics and a synonym dictionary, `GENELIEX` does not require any linguistic resource. To ensure that the improvement brought by `GENELIEX` is not only due to BERT, we have updated the method in [32] with a BERT language model (see Appendix B.7 for details and Table 7 for hyperparameters).
**Evaluation Metrics.** Automatic evaluation of generative models for text is still an open research problem. We rely both on perceptive evaluation and automatic measures to evaluate our model through four criteria (**C1**, **C2**, **C3**,**C4**). **C1** measures Cohesion [17] (*Are the generated sequences grammatically and semantically consistent?*). **C2** (named Sent. in Table 3) evaluates label conservation (*Does the expressed sentiment in the generated sequence match the sentiment of the input sequence?*). **C3** measures the diversity [35] (corresponding to dist1 or dist2 in Table 3[2]) of the sequences (*Does the augmented dataset contain diverse sequences?*). Augmenting the training set with very diverse sequences can lead to better classification performance. **C4** measures the improvement in terms of F1 score when training a classifier (fastText [31]) on the augmented training set (*Does the augmented dataset improve classification performance?*).
**Datasets.** `GENELIEX` is evaluated on two datasets, a medium and a large one (see [50]) which respectively contains 1k and 10k labeled samples. In both cases, we have access to $\mathcal{D}_{g_n}$ a dataset of 80k unlabeled samples. Datasets are randomly sampled from *Amazon* and *Yelp*.
**Experiment description.** We augment extreme regions of each dataset according to three algorithms: `GENELIEX` (with scaling factor $\lambda$ ranging from 1 to 1.5), Kobayashi [32], and Wei and Zou [56]. For each train set's sequence considered as extreme, 10 new sequences are generated using each algorithm. Appendix B.7 gathers further details. For experiment **C4** the test set contains $10^4$ sequences.

### 5.2 Results

**Automatic measures.** The results of **C3** and **C4** evaluation are reported in Table 2. Augmented data with `GENELIEX` are more diverse than the one augmented with Kobayashi [32] and Wei and Zou [56]. The F1-score with dataset augmentation performed by `GENELIEX` outperforms the aforementioned methods on Amazon in medium and large dataset and on Yelp for the medium dataset. It equals state of the art performances on Yelp for the large dataset. As expected, for all three algorithms, the benefits of data augmentation decrease as the original training dataset size increases. Interestingly, we observe a strong correlation between more diverse sequences in the extreme regions and higher F1 score: the more diverse the augmented dataset, the higher the F1 score. More diverse sequences

| Model | Amazon | | | | Yelp | | | |
|---|---|---|---|---|---|---|---|---|
| | Medium | | Large | | Medium | | Large | |
| | F1 | dist1/dist2 | F1 | dist1/dist2 | F1 | dist1/dist2 | F1 | dist1/dist2 |
| Raw Data | 84.0 | X | 93.3 | X | 86.7 | X | 94.1 | X |
| Kobayashi [32] | 85.0 | 0.10/0.47 | 92.9 | 0.14/0.53 | 87.0 | 0.15/0.53 | 94.0 | 0.14/0.58 |
| Wei and Zou [56] | 85.2 | 0.11/0.50 | 93.2 | 0.14/0.54 | 87.0 | 0.15/0.52 | **94.2** | **0.16**/0.59 |
| **GENELIEX** | **86.3** | **0.14/0.52** | **94.0** | **0.18/0.58** | **88.4** | **0.18/0.62** | **94.2** | **0.16/0.60** |

Table 2: Quantitative Evaluation. Algorithms are compared according to **C3** and **C4**. dist1 and dist2 respectively stand for distinct 1 and 2, it measures the diversity of new sequences in terms of unigrams and bigrams. F1 is the F1-score for FastText classifier trained on an augmented labelled training set.

| Model | Amazon | | Yelp | |
|---|---|---|---|---|
| | Sent. | Cohesion | Sent. | Cohesion |
| Raw Data | 83.6 | 78.3 | 80.6 | 0.71 |
| Kobayashi [32] | **80.0** | **84.2** | 82.9 | 0.72 |
| Wei and Zou [56] | 69.0 | 67.4 | 80.0 | 0.60 |
| **GENELIEX** | 78.4 | 73.2 | **85.7** | **0.77** |

Table 3: Qualitative evaluation with three turkers. Sent. stands for sentiment label preservation. The Krippendorff Alpha for Amazon is $\alpha = 0.28$ on the sentiment classification and $\alpha = 0.20$ for cohesion. The Krippendorff Alpha for Yelp is $\alpha = 0.57$ on the sentiment classification and $\alpha = 0.48$ for cohesion.

are thus more likely to lead to better improvement on downstream tasks (*e.g.* classification).
**Perceptive Measures.** To evaluate **C1**, **C2**, three turkers were asked to annotate the cohesion and the sentiment of 100 generated sequences for each algorithm and for the raw data. F1 scores of this evaluation are reported in Table 3. Grammar evaluation confirms the findings of [56] showing that random swaps and deletions do not always maintain the cohesion of the sequence. In contrast, **GENELIEX** and Kobayashi [32], using vectorial representations, produce more coherent sequences. Concerning sentiment label preservation, on Yelp, **GENELIEX** achieves the highest score which confirms the observed improvement reported in Table 2. On Amazon, turker annotations with data from **GENELIEX** obtain a lower F1-score than from Kobayashi [32]. This does not correlate with results in Table 2 and may be explained by a lower Krippendorff Alpha[3] on Amazon ($\alpha = 0.20$) than on Yelp ($\alpha = 0.57$).

# 6 Broader Impact

In this work, we propose a method resulting in heavy-tailed text embeddings. As we make no assumption on the nature of the input data, the suggested method is not limited to textual data and can be extended to any type of modality (*e.g.* audio, video, images). A classifier, trained on aforementioned embedding is dilation invariant (see Equation 1) on the extreme region. A dilation invariant classifier enables better generalization for new samples falling out of the training envelop. For critical application ranging from web content filtering (*e.g.* spam [27], hate speech detection [18], fake news [43]) to medical case reports to court decisions it is crucial to build classifiers with lower generalization error. The scale invariance property can also be exploited to automatically augment a small dataset on its extreme region. For application where data collection requires a huge effort both in time and cost (*e.g.* industrial factory design, classification for rare language [4]), beyond industrial aspect, active learning problems involving heavy-tailed data may highly benefit from our data augmentation approach.

# 7 Acknowledgement

Anne Sabourin was partly supported by the Chaire *Stress testing* from Ecole Polytechnique and BNP Paribas. Concerning Eric Gaussier, this project partly fits within the MIAI project (ANR-19-P3IA-0003).

## Footnotes

[2]dist $n$ is obtained by calculating the number of distinct $n$-grams divided by the total number of generated tokens to avoid favoring long sequences.

[3]measure of inter-rater reliability in $[0, 1]$: 0 is perfect disagreement and 1 is perfect agreement.

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
