[Supplementary Material]

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

# A    Models

## A.1    Background on Adversarial Learning

Adversarial networks, introduced in [25], form a system where two neural networks are competing. A first model $G$, called the generator, generates samples as close as possible to the input dataset. A second model $D$, called the discriminator, aims at distinguishing samples produced by the generator from the input dataset. The goal of the generator is to maximize the probability of the discriminator making a mistake. Hence, if $P_{\text{input}}$ is the distribution of the input dataset then the adversarial network intends to minimize the distance (as measured by the Jensen-Shannon divergence) between the distribution of the generated data $P_G$ and $P_{\text{input}}$. In short, the problem is a minmax game with value function $V(D, G)$

$$\min_{G} \max_{D} V(D, G) = \mathbb{E}_{x \sim P_{\text{input}}}[\log D(x)] + \mathbb{E}_{z \sim P_G}[\log \left(1 - D(G(z))\right)].$$

Auto-encoders and derivatives [26, 34, 21] form a subclass of neural networks whose purpose is to build a suitable representation by learning encoding and decoding functions which capture the core properties of the input data. An adversarial auto-encoder (see [39]) is a specific kind of auto-encoders where the encoder plays the role of the generator of an adversarial network. Thus the latent code is forced to follow a given distribution while containing information relevant to reconstructing the input. In the remaining of this paper, a similar adversarial encoder constrains the encoded representation to be heavy-tailed.

## A.2    Models Overview

Figure 4 provides an overview of the different algorithms proposed in the paper. Figure 4a describes the pipeline for **LHTR** detailed in Algorithm 1. Figure 4b describes the pipeline for the comparative baseline **LHTR**$_1$ where $C^{\text{ext}} = C^{\text{bulk}}$. Figure 4c illustrates the pipeline for the baseline classifier trained on BERT. Figure 4d describes **GENELIEX** described in Algorithm 2, note that the hatched components are inherited from **LHTR** and are not used in the workflow.

Figure 4: Illustrative pipelines.

## A.3    **LHTR** and **GENELIEX** algorithm

This subsection provides detailed algorithm for both models **LHTR** and **GENELIEX**.

**Algorithm 1** LHTR

---

**INPUT:** Weighting coef. $\rho_1, \rho_2, \rho_3 > 0$, Training dataset $\mathcal{D}_n = \{(X_1, Y_1), \ldots, (X_n, Y_n)\}$, batch size $m$, proportion of extremes $\kappa$, heavy tailed prior $P_Z$.
**Initialization:** parameters $(\tau, \theta, \theta', \gamma)$ of the encoder $\varphi_\tau$, classifiers $C_\theta^{ext}, C_{\theta'}^{bulk}$ and discriminator $D_\gamma$
**Optimization:**
  **while** $(\tau, \theta, \theta', \gamma)$ not converged **do**
  Sample $\{(X_1, Y_1) \ldots, (X_m, Y_m)\}$ from $\mathcal{D}_n$ and define $\tilde{Z}_i = \varphi(X_i)$, $i \leq m$.
  Sample $\{Z_1, \ldots, Z_m\}$ from the prior $P_Z$.
  Update $\gamma$ by ascending:

$$\frac{\rho_3}{m} \sum_{i=1}^{m} \log D_\gamma(Z_i) + \log(1 - D_\gamma(\tilde{Z}_i)).$$

  Sort $\{\tilde{Z}_i\}_{i \in \{1, \ldots, m\}}$ by decreasing order of magnitude $||\tilde{Z}_{(1)}|| \geq \ldots \geq ||\tilde{Z}_{(m)}||$.
  Update $\theta$ by descending:

$$\mathcal{L}^{\text{ext}}(\theta, \tau) \stackrel{\text{def}}{=} \frac{\rho_1}{\lfloor \kappa m \rfloor} \sum_{i=1}^{\lfloor \kappa m \rfloor} \ell\big(Y_{(i)}, C_\theta^{\text{ext}}(\tilde{Z}_{(i)})\big).$$

  Update $\theta'$ by descending:

$$\mathcal{L}^{\text{bulk}}(\theta', \tau) \stackrel{\text{def}}{=} \frac{\rho_2}{m - \lfloor \kappa m \rfloor} \sum_{i=\lfloor \kappa m \rfloor + 1}^{m} \ell\big(Y_{(i)}, C_{\theta'}^{\text{bulk}}(\tilde{Z}_{(i)})\big).$$

  Update $\tau$ by descending:

$$\frac{1}{m} \sum_{i=1}^{m} -\rho_3 \log D_\gamma(\tilde{Z}_i) + \mathcal{L}^{\text{ext}}(\theta, \tau) + \mathcal{L}^{\text{bulk}}(\theta', \tau).$$

  **end while**
  Compute $\{\tilde{Z}_i\}_{i \in \{1, \ldots, n\}} = \varphi(X_i)_{i \in \{1, \ldots, n\}}$
  Sort $\{\tilde{Z}_i\}_{i \in \{1, \ldots, n\}}$ by decreasing order of magnitude $||\tilde{Z}_{(1)}|| \geq \ldots ||\tilde{Z}_{(\lfloor \kappa n \rfloor)}|| \geq \ldots \geq ||\tilde{Z}_{(n)}||$.
**OUTPUT:** encoder $\varphi$, classifiers $C^{\text{ext}}$ for $\{x : ||\varphi(x)|| \geq t := ||\tilde{Z}_{(\lfloor \kappa n \rfloor)}||\}$ and $C^{\text{bulk}}$ on the complementary set.

---

**Algorithm 2** GENELIEX: training step

---

**INPUT:** input of LHTR, $\mathcal{D}_{g_n} = \{U_1, \ldots, U_n\}$
**Initialization:** parameters of $\varphi_\tau, C_\theta^{\text{ext}}, C_{\theta'}^{\text{bulk}}, D_\gamma$ and decoder $G_\psi^{\text{ext}}$
**Optimization:**
  $\varphi, C^{\text{ext}}, C^{\text{bulk}} = \text{LHTR}(\rho_1, \rho_2, \rho_3, \mathcal{D}_n, \kappa, m)$
  **while** $\psi$ not converged **do**
  Sample $\{U_1 \ldots, U_m\}$ from the training set $\mathcal{D}_{g_n}$ and define $\tilde{Z}_i = \varphi(X_{U,i})$ for $i \in \{1, \ldots, m\}$.
  Sort $\{\tilde{Z}_i\}_{i \in \{1, \ldots, m\}}$ by decreasing order of magnitude $||\tilde{Z}_{(1)}|| \geq \ldots \geq ||\tilde{Z}_{(m)}||$.
  Update $\psi$ by descending:

$$\mathcal{L}_g^{\text{ext}}(\psi) \stackrel{\text{def}}{=} \frac{\rho_1}{\lfloor \kappa m \rfloor} \sum_{i=1}^{\lfloor \kappa m \rfloor} \ell_{gen.}\big(U_{(i)}, G_\psi^{\text{ext}}(\tilde{Z}_{(i)})\big).$$

  **end while**
  Compute $\{\tilde{Z}_i\}_{i \in \{1, \ldots, n\}} = \varphi(X_i)_{i \in \{1, \ldots, n\}}$
  Sort $\{\tilde{Z}_i\}_{i \in \{1, \ldots, n\}}$ by decreasing order of magnitude $||\tilde{Z}_{(1)}|| \geq \ldots ||\tilde{Z}_{(k)}|| \geq \ldots \geq ||\tilde{Z}_{(n)}||$.
**OUTPUT:** encoder $\varphi$, decoder $G^{\text{ext}}$ applicable on the region $\{x : ||\varphi(x)|| \geq ||\tilde{Z}_{(\lfloor \kappa n \rfloor)}||\}$

---

# B Extreme Value Analysis: additional material

## B.1 Choice of k

To the best of our knowledge, selection of $k$ in extreme value analysis (in particular in Algorithm 1 and Algorithm 2) is still a vivid problem in EVT for which no absolute answer exists. As $k$ gets large the number of extreme points increases including samples which are not large enough and deviates from the asymptotic distribution of extremes. Smaller values of $k$ increase the variance of the classifier/generator. This bias-variance trade-off is beyond the scope of this paper.

## B.2 Preliminary standardization for selecting extreme samples

In Figure 2b selecting the extreme samples on the input space is not a straightforward step as the two components of the vector are not on the same scale, componentwise standardisation is a natural and necessary preliminary step. Following common practice in multivariate extreme value analysis it was decided to standardise the input data $(X_i)_{i \in \{1,\dots,n\}}$ by applying the rank-transformation:

$$\widehat{T}(x) = \left( 1 / \left( 1 - \widehat{F}_j(x) \right) \right)_{j=1,\dots,d}$$

for all $x = (x^1, \dots, x^d) \in \mathbb{R}^d$ where $\widehat{F}_j(x) \stackrel{\text{def}}{=} \frac{1}{n+1} \sum_{i=1}^{n} \mathbb{1}\{X_i^j \leq x\}$ is the $j^{th}$ empirical marginal distribution. Denoting by $V_i$ the standardized variables, $\forall i \in \{1, \dots, n\}, V_i = \widehat{T}(X_i)$. The marginal distributions of $V_i$ are well approximated by standard Pareto distribution, the approximation error comes from the fact that the empirical *c.d.f*'s are used in $\widehat{T}$ instead of the genuine marginal *c.d.f.*'s $F_j$. After this standardization step, the selected extreme samples are $\{V_i, \|V_i\| \geq V_{(\lfloor \kappa n \rfloor)}\}$.

## B.3 Enforcing regularity assumptions in Theorem 1

The methodology in the present paper consists in learning a representation $Z$ for text data *via* `LHTR` satisfying the regular variation condition (2). This condition is weaker than the assumptions from Theorem 1 for two reasons: first, it does not imply that each class (conditionally to the label $Y$) is regularly varying, only that the distribution of $Z$ (unconditionally to the label) is. Second, in Jalalzai et al. [30], it is additionally required that the regression function $\eta(z) = \mathbb{P}\{Y = +1 \mid Z = z\}$ converges uniformly as $\|z\| \to \infty$. Getting into details, one needs to introduce a limit random pair $(Z_\infty, Y_\infty)$ which distribution is the limit of $\mathbb{P}\{Y = \cdot, t^{-1}Z \in \cdot \mid \|Z\| > t\}$ as $t \to \infty$. Denote by $\eta_\infty$ the limiting regression function, $\eta_\infty(z) = \mathbb{P}\{Y_\infty = +1 \mid Z_\infty = z\}$. The required assumption is that

$$\sup_{\{z \in \mathbb{R}_+^d : \|z\| > t\}} \left| \eta(z) - \eta_\infty(z) \right| \xrightarrow[t \to \infty]{} 0. \tag{4}$$

Uniform convergence (4) is not enforced in `LHTR` and the question of how to enforce it together with regular variation of each class separately remains open. However, our experiments in sections 4 and 5 demonstrate that enforcing Condition (2) is enough for our purposes, namely improved classification and label preserving data augmentation.

## B.4 Logistic distribution

The logistic distribution with dependence parameter $\delta \in (0, 1]$ is defined in $\mathbb{R}^d$ by its *c.d.f.* $F(x) = \exp\left\{ - \left( \sum_{j=1}^{d} x^{(j)\frac{1}{\delta}} \right)^\delta \right\}$. Samples from the logistic distribution can be simulated according to the algorithm proposed in Stephenson [51]. Figure 5 illustrates this distribution with various values of $\delta$. Values of $\delta$ close to 1 yield non concomitant extremes, *i.e.* the probability of a simultaneous excess of a high threshold by more than one vector component is negligible. Conversely, for small values of $\delta$, extreme values tend to occur simultaneously. These two distinct tail dependence structures are respectively called 'asymptotic independence' and 'asymptotic dependence' in the EVT terminology.

## B.5 Scale invariance comparison of BERT and `LHTR`

In this section, we compare `LHTR` and BERT and show that the latter is not scale invariant. For this preliminary experiment we rely on labeled fractions of both *Amazon* and *Yelp* datasets respectively

(a) near tail independence     (b) moderate tail dependence     (c) high tail dependence

Figure 5: Illustration of the distribution of the angle $\Theta(X)$ obtained with bivariate samples $X$ generated from a logistic model with different coefficients of dependence ranging from near asymptotic independence Figure 5a ($\delta = 0.9$) to high asymptotic dependence Figure 5c ($\delta = 0.1$) including moderate dependence Figure 5b ($\delta = 0.5$). Non extreme samples are plotted in gray, extreme samples are plotted in black and the angles $\Theta(X)$ (extreme samples projected on the sup norm sphere) are plotted in red. Note that not all extremes are shown since the plot was truncated for a better visualization. However all projections on the sphere are shown.

denoted as *Amazon small dataset* and *Yelp small dataset* detailed in [33], each of them containing 1000 sequences from the large dataset. Both datasets are divided at random in a train set $\mathcal{T}_{\text{train}}$ and $\mathcal{T}_{\text{test}}$. The train set represents ¾ of the whole dataset while the remaining samples represent the test set. We use the hyperparameters reported in Table 4.

|  | NN model | **LHTR$_1$** | **LHTR** |
|---|---|---|---|
| Sizes of the layers $\varphi$ | [768,384,200,50,8,1] | [768,384,200,100] | [768,384,200,150] |
| Sizes of the layers $C_{\theta'}^{bulk}$ | X | [100,50,8,1] | [150,75,8,1] |
| Sizes of the layers $C_{\theta}^{ext}$ | X | X | [150,75,8,1] |
| $\rho_3$ | X | X | 0.001 |

Table 4: Network architectures for *Amazon small dataset* and *Yelp small dataset* . The weight decay is set to $10^5$, the learning rate is set to $5 * 10^{-4}$, the number of epochs is set to 500 and the batch size is set to 64.

**BERT is not regularly varying.** In order to show that $X$ is not regularly varying, independence between $\|X\|$ and a margin of $\Theta(X)$ can be tested [14], which is easily done *via* correlation tests. Pearson correlation tests were run on the extreme samples of BERT and **LHTR** embeddings of *Amazon small dataset* and *Yelp small dataset*. The statistical tests were performed between all margins of $\left(\Theta(X_i)\right)_{1 \geq i \geq n}$ and $\left(\|X_i\|\right)_{1 \geq i \geq n}$.

(a) *Yelp small dataset* - BERT

(b) *Amazon small dataset* - BERT

(c) *Yelp small dataset* - **LHTR**

(d) *Amazon small dataset* - **LHTR**

Figure 6: Histograms of the $p$-values for the non-correlation test between $\left(\Theta(X_i)\right)_{1 \geq i \geq n}$ and $\left(\|X_i\|\right)_{1 \geq i \geq n}$ on embeddings provided by BERT (Figure 6a and Figure 6b) or **LHTR** (Figure 6c and Figure 6d).

Each histogram in Figure 6 displays the distribution of the $p$-values of the correlation tests between the margins $X_j$ and the angle $\Theta(X)$ for $j \in \{1, \ldots d\}$, in a given representation (BERT or **LHTR**) for a given dataset. For both *Amazon small dataset* and *Yelp small dataset* the distribution of the $p$-values is shifted towards larger values in the representation of **LHTR** than in BERT, which means that the correlations are weaker in the former representation than in the latter. This phenomenon is more pronounced with *Yelp small dataset* than with *Amazon small dataset*. Thus, in BERT representation, even the largest data points exhibit a non negligible correlation between the radius and the angle and the regular variation condition does not seem to be satisfied. As a consequence, in a classification setup such as binary sentiment analysis detailed in Section 4.2), classifiers trained on BERT embedding are not guaranteed to be scale invariant. In other words for a representation $X$ of a sequence $U$ with a given label $Y$, the predicted label $g(\lambda X)$ is not necessarily constant for varying values of $\lambda \geq 1$. Figure 7 illustrates this fact on a particular example taken from *Yelp small dataset*. The color (white or black respectively) indicates the predicted class (respectively $-1$ and $+1$). For values of $\lambda$ close to 1, the predicted class is $-1$ but the prediction shifts to class $+1$ for larger values of $\lambda$.

Figure 7: Lack of scale invariance of the classifier trained on BERT: evolution of the predicted label $g(\lambda X)$ from $-1$ to $+1$ for increasing values of $\lambda$, for one particular example $X$.

**Scale invariance of LHTR.** We provide here experimental evidence that **LHTR**'s classifier $g^{\text{ext}}$ is scale invariant (as defined in Equation (1)). Figure 8 displays the predictions $g^{\text{ext}}(\lambda Z_i)$ for increasing values of the scale factor $\lambda \geq 1$ and $Z_i$ belonging to $\mathcal{T}_{\text{test}}$, the set of samples considered as extreme in

the learnt representation. For any such sample $Z$, the predicted label remains constant as $\lambda$ varies, *i.e.* it is scale invariant, $g^{\text{ext}}(\lambda Z) = g^{\text{ext}}(Z)$, for all $\lambda \geq 1$.

(a) *Amazon small dataset*      (b) *Yelp small dataset*

Figure 8: Scale invariance of $g^{\text{ext}}$ trained on LHTR: evolution of the predicted label $g^{\text{ext}}(\lambda Z_i)$ (white or black for $-1/+1$) for increasing values of $\lambda$, for samples $Z_i$ from the extreme test set $\mathcal{T}_{\text{test}}$ from *Amazon small dataset* (Figure 8a) and *Yelp small dataset* (Figure 8b).

## B.6 Experimental settings (Classification): additional details

**Toy example.** For the toy example, we generate 3000 points distributed as a mixture of two normal distributions in dimension two. For training **LHTR**, the number of epochs is set to 100 with a dropout rate equal to 0.4, a batch size of 64 and a learning rate of $5 * 10^{-4}$. The weight parameter $\rho_3$ in the loss function (Jensen-Shannon divergence from the target) is set to $10^{-3}$. Each component $\varphi$, $C^{\text{bulk}}$ and $C^{\text{ext}}$ is made of 3 fully connected layers, the sizes of which are reported in Table 5.

**Datasets.** For Amazon, we work with the video games subdataset from `http://jmcauley.ucsd.edu/data/amazon/`. For Yelp [58, 36], we work with 1,450,000 reviews after that can be found at `https://www.yelp.com/dataset`.

| | Layers' sizes |
|---|---|
| $\varphi$ | [2,4,2] |
| $C^{bulk}_{\theta'}$ | [2,8,1] |
| $C^{ext}_{\theta}$ | [2,8,1] |

Table 5: Sizes of the successive layers in each component of **LHTR** used in the toy example.

**BERT representation for text data.** We use BERT pretrained models and code from the library *Transformers* [4]. All models were implemented using Pytorch and trained on a single Nvidia P100. The output of BERT is a $\mathbb{R}^{768}$ vector. All parameters of the models have been selected using the same grid search.

**Network architectures.** Tables 6 report the architectures (layers sizes) chosen for each component of the three algorithms considered for performance comparison (Section 4), respectively for the moderate and large datasets used in our experiments. We set $\rho_1 = (1 - \hat{\mathbb{P}}(||Z|| \geq ||Z_{(\lfloor \kappa n \rfloor)}||))^{-1}$ and $\rho_2 = \hat{\mathbb{P}}(||Z|| \geq ||Z_{(\lfloor \kappa n \rfloor)}||)^{-1}$.

| | NN model | LHTR$_1$ | LHTR |
|---|---|---|---|
| Sizes of the layers $\varphi$ | [768,384,200,50,8,1] | [768,384,200,100] | [768,384,200,150] |
| Sizes of the layers of $C^{bulk}_{\theta'}$ | [150,75,8,1] | [100,50,8,1] | [150,75,8,1] |
| Sizes of the layers of $C^{ext}_{\theta}$ | X | X | [150,75,8,1] |
| $\rho_3$ | X | X | 0.01 |

Table 6: Network architectures for *Amazon dataset* and *Yelp dataset*. The weight decay is set to $10^5$, the learning rate is set to $1 * 10^{-4}$, the number of epochs is set to 500 and the batch size is set to 256.

## B.7 Experiments for data generation

### B.7.1 Experimental setting

As mentioned in Section 5.1, hyperparameters for dataset augmentation are detailed in Table 7. For

|  | **LHTR** |
|---|---|
| Sizes of the layers $\varphi$ | [768,384,200,150] |
| Sizes of the layers of $C_{\theta'}^{bulk}$ | [150,75,8,1] |
| Sizes of the layers of $C_{\theta}^{ext}$ | [150,75,8,1] |
| $\rho_3$ | 0.01 |

Table 7: For *Amazon* and *Yelp*, we follow [57] the weight decay is set to $10^5$, the learning rate is set to $1 * 10^{-4}$, the number of epochs is set to 100 and the batch size is set to 256.

the Transformer Decoder we use 2 layers with 8 heads, the dimension of the key and value is set to 64 [54] and the inner dimension is set to 512. The architectures for the models proposed by Wei and Zou [56] and Kobayashi [32] are chosen according to the original papers. For a fair comparison with Kobayashi [32], we update the language model with a BERT model, the labels are embedded in $\mathbb{R}^{10}$ and fed to a single MLP layer (this dimension is chosen using the same procedure as in [15, 20]). The new model is trained using AdamW [37].

### B.7.2 Influence of the scaling factor on the linguistic content

Table 8 gathers some extreme sequences generated by **GENELIEX** for $\lambda$ ranging from 1 to 1.5. No major linguistic change appears when $\lambda$ varies. The generated sequences are grammatically correct and share the same polarity (positive or negative sentiment) as the input sequence. Note that for greater values of $\lambda$, a repetition phenomenon appears. The resulting sequences keep the label and polarity of the input sequence but repeat some words [28].

## C   Extremes in Text

**Aim of the experiments**   The aim of this section is double: first, to provide some intuition on what characterizes sequences falling in the extreme region of **LHTR**. Second, to investigate the hypothesis that extremes from **LHTR** are input sequences which tend to be harder to model than non extreme ones

Regarding the first aim ( *(i) Are there interpretable text features correlated with the extreme nature of a text sample?*, since we characterize extremes by their norm in **LHTR** representation, in practice the question boils down to finding text features which are positively correlated with the norm of the text samples in **LHTR**, which we denote by $\|\varphi(X)\|$ and referred to as the '**LHTR** norm' in the sequel. Preliminary investigations did not reveal semantic features (related to the meaning or the sentiment expressed in the sequence ) displaying such correlation. However we have identified two features which are positively correlated both together and with the norm in **LHTR**, namely the sequence length $|U|$ as measured by the number of tokens of the input (recall that in our case an input sequence $U$ is a review composed of multiple sequences ), and the norm of the input in BERT representation ('BERT norm', denoted by $\|X\|$).

As for the second question ( *(ii) Are **LHTR**'s extremes harder to model?* ) we consider the next token prediction loss [5] ('LM loss' in the sequel) obtained by training a language model on top of BERT. The next token prediction loss can be seen as a measure of hardness to model the input sequence. The question is thus to determine whether this prediction loss is correlated with the norm in **LHTR** (or in BERT, or with the sequence length).

**Results**   Figure 9 displays pairwise scatterplots for the four considered variables on *Yelp* dataset (left) and *Amazon* dataset (right). These scatterplot suggest strong dependence for all pairs of variables. For a more quantitative assessment, Figure 10 displays the correlation matrices between the four quantities $\|\varphi(X)\|$, $\|X\|$, $|U|$ and 'LM Loss' described above on Amazon and Yelp datasets. Pearson and Spearman two-sided correlation tests are performed on all pairs of variables, both tests having as null hypothesis that the correlation between two variables is zero. For all tests, $p$-values are smaller than $10^{-16}$, therefore null hypotheses are rejected for all pairs.

These results prove that the four considered variables are indeed significantly positively correlated, which answers questions $(i)$ and $(ii)$ above.

Figure 9: Scatterplots of the four variables 'BERT norm', '`LHTR` norm', 'LM loss' and 'sequence length' on *Yelp* dataset (top) and *Amazon* dataset (bottom).

| | |
|---|---|
| Input | very sloppy and slow service. when we arrived they told us to sit anywhere but all the tables were still dirty and haven't been cleaned. they didn't bother to ask if we wanted refills on our drinks. we needed an extra plate and didn't get one so my nephew decides to go up to the counter and ask for one because he's hungry. they gave our check when we were still eating. the list can go on and on. i wouldn't recommend this place. go somewhere else for faster and better service. very disappointed |
| $\lambda = 1.1$ | very sloppy and sluggish service. when we got there, they told us to sit anywhere but all the tables were empyt full of dishes and were not cleaned at all. they didn't bother to ask if our drinks would be added. we needed an extra dish and didn't get one, so my cousin decided to go to the counter and ask one because he's hungry. they were going to watch while we were still eating. the list could go on and on. i would not recommend this place. go elsewhere for faster and better service. very very disappointed |
| $\lambda = 1.2$ | services and survivors. when he got there, he told us we were sitting everywhere but all the tables were full of dishes and we didn't wash everything. he never bothered to ask if our drinks would be added. we needed extra food and didn't get one, so my brother decided to go to the locker and ask because he was thirsty. they want to watch it while we eat. the list can be continuous and active. i would not recommend this place. go elsewhere for faster and better service. very disappointed |
| $\lambda = 1.3$ | services and survivors. when he got there, he told us that we were sitting everywhere, but all the tables were full of dishes and we didn't wash everything. he never bothered to ask if our drinks would be added. We needed more food and we didn't get it, so my brother decided to go to the locker and ask because he was thirsty. they want to watch it when we eat. the list can be continuous and active. i would not recommend this place. go faster and faster for better service. very disappointed |
| Input | visited today with my husband. we were in the firearms section. there were 3 employees in attendance with one customer. my husband ask a question and was ignored. he waited around for another 10 minutes or so. if it had been busy i could understand not receiving help. we left and went elsewhere for our purchases. |
| $\lambda = 1.1$ | visited today with my husband. we were in the firearms section. together with one customer there were 3 employees. my husband asked and was ignored. waited about another 10 minutes. if it was busy, i would understand that i wouldn't get help. we left and went somewhere else because of our purchases. |
| $\lambda = 1.2$ | today she visited with her husband. we were in the gun department. there were 3 employees together with one customer. my husband asked and was ignored. waited another 10 minutes. if he was busy, i would understand that i would not receive help. we went and went somewhere else because of our shopping. |
| $\lambda = 1.3$ | today, she went with her husband. we are in the gun department. there are 3 employees and one customer. my husband rejected me and ignored him. wait another minute. if he has a job at hand, i will understand that i will not get help. we went somewhere else because of our business. |
| Input | walked in on a friday and got right in. it was exactly what i expected for a thai massage. the man did a terrific job. he was very skilled, working on the parts of my body with the most tension and adjusting pressure as i needed throughout the massage. i walked out feeling fantastic and google eyed. |
| $\lambda = 1.1$ | walked in on a friday and got right in. it was exactly what i expected for a thai massage. the man did a terrific job. he was very skilled, working on the parts of my body with the most tension and adjusting pressure as needed throughout the massage. i walked out feeling fantastic and google eyed. |
| $\lambda = 1.2$ | climb up the stairs and get in. the event that i was expecting a thai massage. the man did a wonderful job. he was very skilled, dealing with a lot of stress and stress on my body parts. i walked out feeling lightly happy and tired. |
| $\lambda = 1.3$ | go up and up. this was the event i was expecting a thai massage. the man did a wonderful job. what this was was an expert, with a lot of stress and stress on my body parts. i walked out feeling lightly happy and tired. |
| Input | i came here four times during a 3 - day stay in madison. the first two was while i was working - from - home. this place is awesome to plug in, work away at a table, and enjoy a great variety of coffee. the other two times, i brought people who wanted good coffee, and this place delivered. awesome atmosphere. awesome awesome awesome. |
| $\lambda = 1.1$ | i came here four times during a 3-day stay in henderson. the first two were while i was working - from home. this place is great for hanging out, working at tables and enjoying the best variety of coffee. the other two times, i brought in people who wanted a good coffee, and it delivered a place. better environment. really awesome awesome. |
| $\lambda = 1.2$ | i came here four times during my 3 days in the city of henderson. the first two were while i was working - at home. this place is great for trying, working tables and enjoying the best variety of coffee. the other two times, i brought people who wanted good coffee, and it brought me somewhere. good environment. really amazing. |
| $\lambda = 1.3$ | i came here four times during my 3 days in the city of henderson. the first two are when i'm working - at home. this place is great for trying, working tables and enjoying a variety of the best coffees. the other two times, i bring people who want good coffee, and that brings me somewhere. good environment. very amazing. |

Table 8: Sequences generated by **GENELIEX** for extreme embeddings implying label (sentiment polarity) invariance for generated Sequence. $\lambda$ is the scale factor. Two first reviews are negatives, two last reviews are positive.

Figure 11 provides additional insight about the magnitude of the shift in sequence length between extremes in the **LHTR** representation and non extreme samples. Even though the histograms overlap (so that two different sequences of same length may be regarded as extreme or not depending on other factors that are not understood yet), there is a visible shift in distribution for both *Yelp* and *Amazon* datasets, both for the positive and negative class in the classification framework for sentiment analysis. Kolmogorov-Smirnoff tests between the length distributions of the two considered classes for each label were performed, which allows us to reject the null hypothesis of equality between distributions, as the maximum $p$-values is less than 0.05.

Figure 10: Non diagonal entries of the correlation matrices of the four variables 'BERT norm', '`LHTR` norm', 'LM loss' and 'sequence length' for *Yelp* dataset (left) and *Amazon* dataset (right).

(a) *Yelp* - labeled $+1$

(b) *Yelp* - labeled $-1$

(c) *Amazon* - labeled $+1$

(d) *Amazon* - labeled $-1$

Figure 11: Histograms of the samples' sequence length for *Yelp* dataset (Figure 11a and Figure 11b) and *Amazon* (Figure 11c and Figure 11d). The number of sequences in the bulk is approximately 3 times the number of extreme sequences for each dataset 10000 sequences are considered and extreme region contains approximately 3000 sequences .

**Experimental conclusions**    We summarize the empirical findings of this section:

1. An 'extreme' text sequence in `LHTR` representation is more likely to have a greater length (number of tokens) than a non extreme one.

2. Positive correlation between the BERT norm and the `LHTR` norm indicates that a large sample in the BERT representation is likely to have a large norm in the `LHTR` representation as well: the learnt representation `LHTR` taking BERT as input keeps invariant (in probability) the ordering implied by the norm.

3. A consequence of the two above points is that long sequences tend to have a large norm in BERT.

4. Extreme text samples (regarding the BERT norm or the `LHTR` norm) tend to be harder to model than non-extreme ones.

5. Since extreme texts are harder to model and also somewhat harder to classify in view of the BERT classification scores reported in Table 1, there is room for improvement in their analysis and it is no wonder that a method dedicated to extremes *i.e.* relying on EVT such as `LHTR` outperforms the baseline.