[Reviews · NeurIPS 2020]

Review 1

Summary and Contributions: This paper proposes a method for learning heavy-tailed embeddings that can be used in principled frameworks for generalization in extreme regions of the input space. In terms of applications, the paper is relevant and timely, as building robust classifiers for general embedding-based architectures (like BERT) is of growing interest. The two main contributions of the paper are: 1) an algorithm for learning heavy-tailed representations so that the resulting inputs satisfy the regular variation assumptions for the extreme value classification. 2) an algorithm for data augmentation in the extreme value region (without relying on external linguistic resources). This is particularly important for applications in which data augmentation helps most in the heavy tail (which is the hardest to get data for). ------- UPDATE: Thanks to the authors for their response! Including a special treatment of extreme vs not examples in Table 2 might be useful to the reader to highlight the impact of your method. I think that, given extra space, "uncramming" sections 4-5 will greatly help readability.

Strengths: In general, I find the paper appealing. The proposed methods are well motivated theoretically, and the empirical results show convincing gains on "extreme regions" of the sentiment tasks (across two datasets). Using the proposed model, which uses a standard classifier for the "bulk" and the extreme region classifier for the "extreme", gets the best of both worlds (as the extreme classifier generally does *worse* on the bulk). It is also encouraging that the data augmentation increases F1 score (by a slight margin, more so on the smaller dataset sizes). Question: are the F1 scores here (Table 2) stratified at all by "extreme" vs not?

Weaknesses: 1) The proposed method regularizes fixed BERT embeddings to be heavy tailed, however most SOTA methods fine-tune the entire BERT model (instead of just an MLP on top). Though it seems like the proposed losses can still regularize these representations (affecting BERT + \phi now instead of just \phi), it's unclear if it will be as effective. It would be good to note whether or not fixing BERT is just for computational efficiency or not. 2) Sentiment analysis is not the most compelling task; it would be good to show the generality of this method on other tasks (and to show that extreme values exist). For tasks in which length doesn't clearly affect decisions (e.g., with more localized predictions such as in NER, relation extraction, or the majority of SQuAD-like QA pairs) such extreme regions might be quite different than the ones focused on here.

Correctness: The claims and empirical methodology are fine.

Clarity: In general, the paper is well written and clear to follow (if a bit scrunched for space). A few minor comments: L11: controllable attributes L58: Seems like an unintended sentence break. L88: whose instead of which, or "of which the main focus" Table 1: Proposed model overall for amazon has a , instead of a . Figure 4 would benefit from higher resolution.

Relation to Prior Work: The framing of this work is fine.

Reproducibility: Yes

Additional Feedback:


Review 2

Summary and Contributions: Empirical risk minimizers perform poorly in extreme regions because very large observations are rare and their contribution to the empirical error is irrelevant. This problem is tackled by the proposed framework for classification in the extreme by Jalalzai et al. (2018). The authors rely on this framework for binary classification to present a data augmentation algorithm and an algorithm to learn heavy-tailed representations for text data. Although the proposed methods are not limited to text data, because the type of input data is not predefined, as the authors clearly point out, this paper focuses on modifying the high dimensional vectors issued by BERT.

Strengths: The evaluation of the presented algorithms is done with two different experiments. On the one hand a binary sentiment classification task based on Yelp and Amazon reviews, on the other hand a text generation task comparing the proposed method to two existing dataset augmentation methods. The results on the later are evaluated through four criteria (e.g. the sentiment of the generated sequences) on two datasets. The evaluation of the classification task compares a baseline model trained on BERT with the classifiers trained by the proposed algorithm LHTR. To ensure the statistical guarantees for classification in extreme regions (Jalalzai et al., 2018), two different classifiers are trained: One for the extreme region where just few training points are available and one for the complementary set which can be replaced by any other classifier trained on the (restricted) input data. On the extreme region (25% of the test set) the special trained classifier performs better than the baseline classifier. The results of the experiments are well documented and seem reasonable.

Weaknesses: Presenting two algorithms (one training an encoder and two classificators and one training an encoder and a decoder) and evaluating them on two tasks while recalling the classification setup for extremes by Jalalzai et al. (2018) the contribution is very comprehensive. Thus, some of the most interesting details and findings only appear in the supplementary material.

Correctness: As far as I can judge the method is correct.

Clarity: The Paper is well written and error free, but the huge extensive use of formulas makes the text in some parts not easy to follow. Probably due to space constraints, a lot of figures and tables are only available in the supplementary material although they contribute a lot to a better understanding of the methods and findings. At least the figures illustrating the proposed algorithms LHTR and GENELIEX should be in the main text.

Relation to Prior Work: The authors clearly point out their algorithm for training the classifiers (LHTR) differs from other encoding schemes by using a heavy-tailed, regularly varying distribution instead of a Gaussian one. Thus the proposed data augmentation algorithm (GENELIEX) also differs from other work, because LHTR is part of its pipeline. In the experiments, LHTR is compared to a baseline model while GENELIEX is compared to two existing data augmentation models.

Reproducibility: Yes

Additional Feedback:


Review 3

Summary and Contributions: This paper attempts to use multivariate extreme value theory to allow better representation of word/values at the tail of a distribution, to learn "heavy-tailed embeddings". They perform experiments on real and synthetic datasets with empirical results of how this affects end-of-distribution words. Update: I think the author response was helpful to clarify certain points. Overall, I think we all agree that this is something that affects *nearly every* NLP task, since words are the fundamental unit that need to be taken in/processed, and currently, the only strategy to allow better performance/computation is to use heuristics to filter out words at the tail end of the distribution. However, those words are often very relevant to the task and there should be measures to take them into account. I also think this data augmentation with scale invariance is a novel enough and different from all the other data augmentation strategies that exist. My score was mildly positive so I'm updating it to be higher---I think this is a good contribution!

Strengths: 1. This is an important problem that is usually skimmed-over in most NLP tasks (e.g,. focusing on just the most frequent/most relevant words in instances with large vocabularies). 2. They introduce a data augmentation method that relies on the scale invariance properties which could potentially be useful for a range of NLP tasks. 3. This paper is well-written and clearly explained; especially the theoretical motivation and how insights from extreme value theory can be built into classifiers that can be fit into tasks of interest. 4. The experimental setup is sound and clearly explained.

Weaknesses: 1. It remains to be seen if the data augmentation is actually helpful for a range of tasks. For most NLP use-cases, it seems like simple heuristics (that albeit are somewhat flawed) that try to look at only most relevant or frequent words, actually improve performance as well as lower training time/cost because of a fewer number of trainable parameters in the embedding layer. 2. ^if authors could comment on/explore if these really does help performance across a range of tasks that would greatly improve the contributions of this work. 3. Several typos/inconsistencies throughout the paper. (e.g., Table 1, "," instead of a decimal point? in Table 3, the bolded numbers are inconsistent?) 4. There are no qualitative examples of outputs---this would especially be useful to see in this setting, for different words at the end of the distribution.

Correctness: The experimental setup is sound and explained in detail.

Clarity: The paper is written well and all experimental details and tables and figures are adequately explained.

Relation to Prior Work: This is well-positioned in the literature; however ties to other data-augmentation papers could be made (although those do not really focus on end-of-distribution words) in the same way.

Reproducibility: Yes

Additional Feedback:


Review 4

Summary and Contributions: This paper presents two tasks informed by extreme value theory. The first (LHTR) is learning a transformation Z of an embedding X to improve downstream classifier performance in fat tailed regions. The next (GENELIEX) is a data augmentation algorithm that operates directly on the embedding and provides a good alternative to the ad-hoc methods that include token-level perturbations. LHTR in particular resembles autoencoder schemes where a fat-tailed distribution is used for Z as opposed to a gaussian. Experiments show that combining a regular classifier for the sample heavy region and LHTR for the fat tailed region provides a performance boost. Human evaluation shows GENELIEX synthesized samples are more coherent (in cases where a higher alpha is observed). POST REBUTTAL Overall a good paper - I am going to keep my (already positive) score as-is. I appreciate the authors' attempts to address my comments and hope to see this paper at NeurIPS.

Strengths: The work is well presented. It exploits the fat tailed nature of text content (vis-a-vis image) and utilizes it in two valuable downstream applications. The text augmentation task in particular is a strong addition and these ideas are foundationally stronger than token perturbation methods we typically see. Overall, a good NLP paper and a good mix of theory and application.

Weaknesses: Empirically there is 1 toy dataset and 2 text data sets explored in this paper (plus BERT embeddings). I would posit that a larger set of corpora be utilized. How does the alpha value impact performance would be good to see here.

Correctness: Yes.

Clarity: Yes.

Relation to Prior Work: Yes.

Reproducibility: Yes

Additional Feedback:

[Author Response · NeurIPS 2020]

**General response.** We thank the reviewers for their helpful comments and remarks. We are glad that you appreciated our efforts for the scientific clarity and the experimental description. We are pleased that you acknowledge the mix between theory and practice in our work.

We note that reviewers are interested in extensions of the current paper to a wider range of tasks and corpora: Reviewer 1 states that *'Sentiment analysis is not the most compelling task; it would be good to show the generality of this method on other tasks'*. Reviewer 3 suggests to assess *'if the data augmentation scheme is actually helpful for a range of tasks'* and Reviewer 4 mentions to work with *'a larger set of corpora'*.
↪ We agree. In this paper, we introduce **LHTR** and **GENELIEX**. We worked on two datasets, *Amazon* with 231k reviews and *Yelp* with 1450k reviews as stated on line 261, and further detailed in Appendix B.6. The focus is here on a method for learning a heavy-tailed representation (including a data-augmentation procedure) and its impact on a standard text classification task, namely sentiment analysis and on data augmentation. Table 2 shows that when the training set is augmented with **GENELIEX**, the classification performance increases (higher F1 score) for both *Amazon* and *Yelp* datasets (both medium and large size). Also, when compared with other methods, we achieve better performance. Future work will be the opportunity to address a wider range of tasks as it will also be the opportunity to work on a larger set of corpora, extending the work presented in this article.

**Reviewer 1.** Thank you for spotting the typos on lines 11, 58, 88 and Table 1. We will update the paper accordingly.

• *'Question: are the F1 scores here (Table 2) stratified at all by "extreme" vs not?'*
↪ A fasttext classifier is trained on the augmented training set (with various methods including **GENELIEX**). Table 2 reports fasttext F1-score computed on the whole test set with no special treatment made for extreme samples. Thus no stratification sampling is required.

• *'The proposed method regularizes fixed BERT embeddings to be heavy tailed, however most SOTA methods fine-tune the entire BERT model (instead of just an MLP on top). Though it seems like the proposed losses can still regularize these representations (affecting BERT + $\phi$ now instead of just $\phi$), it's unclear if it will be as effective. It would be good to note whether or not fixing BERT is just for computational efficiency or not.'*
↪ BERT embedding was fixed for both computational efficiency and for evaluating the improvement solely resulting from $\phi$ in our experiments. We plan on fine-tuning the entire BERT + $\phi$ model during the training phase in future work.

**Reviewer 2.** Upon acceptance, the additional ninth page will be the opportunity to include the figures describing **LHTR** and **GENELIEX** content currently in the supplementary material.

**Reviewer 3.** Thank you for spotting the typos in Table 1. We will update the paper.

• *'There are no qualitative examples of outputs—this would especially be useful to see in this setting, for different words at the end of the distribution.'*
↪ Please refer to Table 8 in Appendix B.7.2. for output examples generated by **GENELIEX**.

**Reviewer 4.** We agree that text augmentation would particularly benefit from our embedding's scale invariance which is foundationally stronger than known token perturbation methods.

• *'How does the alpha value impact performance would be good to see here.'*
↪ The alpha value corresponds to the tail index of Z's heavy-tailed distribution. As the tail index increases, the tail gets lighter. It results that the greater alpha is the less likely it is for extremes to occur. Although we highlight that the approach is generic, in our experiments, the selected distribution is a multivariate Logistic distribution (see l. 220 and Appendix B.4). Other heavy-tailed distributions (with different tail indexes) may be selected.

[Meta-Review · NeurIPS 2020]

This paper proposes a method for learning text embeddings that uses EVT (extreme value theory) to generalize better in extreme regions of the input space. The contribution is sound, meaningful, as well a timely, and I think the paper will be of significant interest to a large portion of the Neurips audience.